# D-Penicillamine-Induced Myasthenia Gravis—A Probable Complication of Wilson’s Disease Treatment—A Case Report and Systematic Review of the Literature

**DOI:** 10.3390/life13081715

**Published:** 2023-08-10

**Authors:** Agnieszka Antos, Anna Członkowska, Jan Bembenek, Iwona Kurkowska-Jastrzębska, Tomasz Litwin

**Affiliations:** 1Second Department of Neurology, Institute of Psychiatry and Neurology, 02-957 Warsaw, Poland; agantos@ipin.edu.pl (A.A.); czlonkow@ipin.edu.pl (A.C.); ikurkowska@ipin.edu.pl (I.K.-J.); 2Department of Clinical Neurophysiology, Institute of Psychiatry and Neurology, 02-957 Warsaw, Poland; jbembenek@ipin.edu.pl

**Keywords:** Wilson’s disease, d-penicillamine, myasthenia gravis, pyridostigmine, antibodies against acetylcholine receptors

## Abstract

Wilson’s disease (WD) is a genetic disorder with copper accumulation in various tissues leading to related clinical symptoms (mainly hepatic and neuropsychiatric) which can be in 85% of patients successfully treated with anti-copper agents. However, during WD treatment neurological deterioration may occur in several patients. D-penicillamine (DPA) is one of the most frequently used drugs in WD treatment. Despite its efficacy, DPA can produce many adverse drug reactions, which should be recognized early. We present the case of a 51-year-old man diagnosed with the hepatic form of WD and initially treated with DPA in whom after 15 months of treatment, diplopia and evening ptosis occurred. WD treatment non-compliance as well as overtreatment were excluded. Supported by neurological symptoms, a positive edrophonium test, and high serum levels of antibodies against acetylcholine receptors (AChR-Abs), as well as low concentrations of antibodies against muscle-specific kinase (MuSK-Abs), the diagnosis of myasthenia gravis (MG), induced by DPA, was established. DPA was stopped; zinc sulfate for WD and pyridostigmine for MG symptoms were introduced. Diplopia and ptosis subsided after a few days, which supported our diagnosis. During a follow-up visit after 6 months, the patient did not present any MG symptoms. AChR-Abs level gradually decreased and MuSK-Abs were no longer detected. Pyridostigmine was stopped, and within 9 months of follow-up, the neurological symptoms of MG did not reoccur. The authors discussed the patient’s neurological deterioration, performed a systematic review of DPA-induced MG in WD and concluded that MG is a rare and usually reversible complication of DPA treatment. DPA-induced MG generally occurs 2–12 months after treatment initiation and ocular symptoms predominate. Response to pyridostigmine treatment is good and MG symptoms usually reverse within one year after DPA treatment cessation. However, symptoms may persist in some cases where DPA treatment is only a trigger factor for MG occurrence.

## 1. Introduction

Wilson’s disease (WD) is an inherited, rare (prevalence 1:30,000) disorder of copper metabolism, with pathological copper accumulation in many organs and tissues and subsequent damage to the affected tissues and related clinical symptoms (mainly hepatic and/or neuropsychiatric) [1,2].

WD is potentially treatable with pharmacological agents including: (1) chelators that increase urinary copper excretion (d-penicillamine [DPA] or trientine) and (2) zinc salts that decrease absorption of copper from the digestive tract [3,4,5]. In general, if detected early and treated correctly, the prognosis is good: almost 85% of patients improve or their symptoms and diagnostic test results stabilize [6]. However, some patients’ neurological deterioration (early or late) may occur [7], as well as WD overtreatment or adverse drug reactions (ADR’s)—associated with anti-copper treatment leading to neurological symptoms aggravation [8]. 

DPA was discovered and introduced as a treatment for WD in 1956 by John Walshe. It has been used for almost 70 years and still remains one of the most frequently used treatment options in WD worldwide [1,2,3]. Current international guidelines for the treatment of WD recommend the use of chelators (including DPA) as the first-choice treatment especially in symptomatic hepatic WD patients [1]. Despite its proven efficacy, DPA is suggested to be a risk factor for early neurological deterioration (in case of too fast introduction of anti-copper treatment) and is known to induce autoimmune response and cause rare but clinically significant autoimmune ADRs including myasthenia gravis (MG), which every physician involved in WD treatment should be aware of [1,2,9,10,11,12,13,14].

MG is a rare autoimmune disorder (prevalence 4:10,000) caused by antibodies (Abs) targeting the postsynaptic muscle end plate [15,16,17]. The most common Abs in MG are against human nicotinic acetylcholine receptors (AChR) or muscle-specific kinase (MuSK), and less often, Abs against low-density lipoprotein receptors-related protein 4 (LRP4), which affects the neuromuscular junctions of skeletal muscles [15,16,17]. The role of AChR-Abs is to bind to extracellular domains of the receptor, damaging signal transduction and the postsynaptic membrane, and also causing receptor crosslinking and delayed receptor internalization [16]. MuSK-Abs can bind to the extracellular domains of AChRs, reduce the number of AChRs, and impair cooperation between the motor nerve ending and the postsynaptic membrane in the neuromuscular junctions [16]. By changing neurotransmission at the postsynaptic part of neuromuscular junctions, both Abs produce painless, fatigable muscle weakness, which is typically more evident after exercise (apocamnosis) [15,16,17]. MG often presents with ocular symptoms (ptosis and diplopia—ocular form), and/or reduction of facial expression, bulbar presentation (speech and swallowing problems), and neck and limb weakness (generalized form) [15,16,17]. As MG is an autoimmune disorder, about 10% of patients with MG are diagnosed with thymoma [15]. Moreover, a higher occurrence of concomitant autoimmune disorders in patients with MG was reported [15,16,17]. Additionally, several medications may induce MG (including DPA, immune checkpoint inhibitors, tyrosine kinase inhibitors, interferons, statins, etc.) or exacerbate its symptoms (including macrolides, aminoglycosides, fluoroquinolones, penicillins, beta-adrenergic blockers, magnesium, antipsychotics, lithium, L-type calcium blockers, etc.) [15,16,17].

Since WD patients may present with a wide spectrum of neurological symptoms, diagnosing DPA-induced MG may be difficult. Additionally, any neurological deterioration in WD needs to be quickly diagnosed and treated according to leading pathology (DPA-related deterioration? non-compliance with anti-copper drugs? WD overtreatment?) [7,8]. As there is little data in the literature about DPA-induced MG, we aimed to present a case of a patient with WD who developed DPA-induced MG and whose symptoms completely subsided after DPA cessation, with an additional systematic review of DPA-induced MG in WD. 

## 2. Materials and Methods

### 2.1. Case Report

A 51-year-old man was diagnosed with WD in March 2021 during differential diagnosis of liver disease. The diagnostic tests of liver injury were started due to increased values of liver function tests (LFTs; ALT 158 U/L [normal < 41]; AST 80 U/L [normal < 40]) performed during routine examinations in outpatient occupational medicine visits. Based on decreased serum ceruloplasmin levels (14.7 mg/dL [normal 20–60]), increased 24 h urinary copper excretion (60 μg/24 h [normal < 50]) and the presence of Kayser–Fleischer ring (confirmed with slit lamp examination), the diagnosis of WD was established. Patient at the diagnosis of WD had no neurological symptoms, there were also no pathological changes in brain magnetic resonance imaging (MRI) performed at that time.

Treatment with DPA was introduced, up to 1000 mg/day. After 15 months of treatment, LFTs normalized; however, double vision when looking down and evening ptosis of both eyelids occurred. The patient was referred to the neurological department. In the neurological examination, typical symptoms of MG were found including diplopia and bilateral fatigable ptosis (apocamnosis) that fluctuated throughout the exam. Brain MRI, brain angio-MRI and routine blood tests including thyroid functions were normal. The copper metabolism analysis documented the correct compliance with anti-copper treatment and excluded copper deficiency (daily urinary copper excretion 480 µg/24 h (the optimal values for WD patients treated with DPA: 200–500 µg/24 h; serum ceruloplasmin 12 mg/dL and serum copper 42 µg/dL [normal 70–140]. Cerebrospinal fluid (CSF) test results were normal and neuroboreliosis was excluded. Repetitive nerve stimulation in the left orbicularis oculi and left abductor pollicis brevis, and single fiber electromyography (the most sensitive diagnostic test, especially for seronegative MG) in the left orbicularis oculi muscle did not reveal signs of abnormal neuromuscular junction transmission. However, the edrophonium test was positive, with high serum levels of AChR-Abs (15.3 nmol/L [normal < 0.45], using an enzyme-linked immunosorbent assay) and the presence of MuSK-Abs (titration 1:20 [negative < 1:16] using an indirect immunofluorescence test involving transfected Hep-2 MuSK cells). Computed tomography of the mediastinum did not confirm the presence of thymoma or hyperplasia. The diagnosis of MG, probably related to DPA, was established (Naranjo score = 4). There are previous reports on such reactions (1 point), the occurrence of ADR after the suspected drug was administered (2 points), ADR improvement after drug discontinuation (1 point). ADR was confirmed by objective evidence (MG Abs and clinical symptoms (1 point) [18].

DPA was stopped, zinc sulphate 180 mg/day was initiated as a WD treatment and pyridostigmine 240 mg/day was introduced to treat the symptoms of MG. Diplopia and ptosis subsided after a few days. At the follow-up visit after 6 months, the patient did not present MG symptoms. In the follow-up electromyography examination, the mean Consecutive Difference of this patient’s single-fiber EMG from orbicularis oculi muscle repeated after the MG symptoms resolved was 23 µs (laboratory norm up to 31 µs), AChR-Abs levels gradually decreased (6.3 nmol/L) and MuSK-Abs were no longer detected. Pyridostigmine was stopped and neurological symptoms did not reoccur until April 2023 (Figure 1). 

In summary, our patient’s MG clinical symptoms, which occurred after DPA introduction and subsided after DPA cessation with a gradual decrease of AChR-Abs and disappearing MuSK-Abs, allowed the probable diagnosis of DPA-induced MG to be established. The patient will be further followed up by our department.

### 2.2. Systematic Review of DPA-Induced MG in WD Patients

This systematic review was performed according to the internationally accepted criteria of the Preferred Reporting Items for Systematic Reviews and Meta-Analyses (PRISMA) statement [19].

We searched the PubMed, Embase, Web of Science and Scopus databases (up to 1 June 2023) for original studies, as well as case and series reports analyzing reports of DPA-induced MG in WD patients. Our search terms were: (“Wilson’s disease”/“Wilson disease” and ”myasthenia”) and (“Wilson’s disease”/“Wilson disease” and ”myasthenia and d-penicillamine”). The eligible studies for further analysis were: (1) published in English; (2) human studies; and (3) original as well as case and series reports describing MG-induced DPA in WD patients. Additionally, reference lists of the publications were searched.

Due to the low number of articles analyzed, all titles and abstracts retrieved in the searches were screened by all authors and duplicate, overlapping and irrelevant reports were removed. The full texts of eligible articles were analyzed by all authors to confirm eligibility and then authors analyzed the data and observations.

Initially, we found 151 records during the PubMed search (Figure 2). After the removal of duplicate and irrelevant articles, 7 records remained [20,21,22,23,24,25,26]. One paper published in 1999 was not available [24] and this was also excluded from the analysis. Finally, 6 papers describing 6 case reports were included (Table 1) [20,21,22,23,25,26]. In an additional search of the Scopus, Embase, and Web of Science databases, we did not find any additional reports. 

DPA-induced MG occurred mostly in women (4/6; 75%). Treatment duration with DPA ranged from 6 to 96 months (mean: 39.8 months). After DPA cessation, MG symptoms disappeared in all cases and patients were switched to another anti-copper drug (mostly trientine). However, in one case, the physicians decided to diagnose MG and treat the patient with pyridostigmine due to persistent increased serum AChR-Ab; DPA treatment was continued and thymectomy was performed [25]. In another case, DPA was reintroduced after MG symptoms regression, without further complications [26].

## 3. Discussion

MG belongs to a group of rare and usually reversible complications of DPA treatment that occur mostly (but not exclusively) in autoimmune disorders [15,16,17]. Based on data from the literature, DPA-induced MG occurs usually after 2–12 months but can be evident years after treatment initiation, especially in WD patients (6–96 months). The symptoms are generally ocular, with good response to pyridostigmine and clinical remission within one year after DPA treatment cessation: 70% of patients completely clinically recover, in contrast to idiopathic MG where 9.6 to 57.5% of patients will have remission during the first year [16,17,18,19,20,21,22,23,24,25,26,27,28,29,30,31]. If MG symptoms persist, it seems that DPA treatment is only a trigger factor for MG. Long-term observations after DPA cessation, AChR-Abs decrease, or even disappearance, as well as lack of clinical symptoms during follow-up, allow us to retrospectively establish the etiology of MG in such cases [15,16,17,25,27].

Data on the incidence of DPA-induced ADR are conflicting. Various papers suggest that 1–7% of patients treated with DPA develop symptoms of MG [15,16,17]. However, mostly in the course of RA treatment and only occasionally in WD [2,15,16,17]. From several retrospective analyses of WD patients treated with anti-copper agents (including DPA), it appears that cases of DPA-induced MG are incidental, with only seven cases in the medical literature and three cases in a Polish cohort of 1000 WD patients (data not published), indicating that it occurs less frequently that previously suggested [15,18,20,21,22,23,24,25,26,27,28,29].

The pathogenesis of DPA-induced MG is not clear, and the exact mechanism remains unknown. An increase in AChR-Abs, which is also seen in idiopathic MG, may suggest a common pathogenesis. The pathogenesis of MG includes binding of DPA to AChR and altering the autoimmunogenicity of this complex, leading to the production of AChR-Abs. It is also suspected that DPA can cause modification of major histocompatibility complex molecules and peptides on the antigen-presenting cell surface, which causes autoimmunity against AChR [32]. Due to the size of the molecule and its reactive sulfhydryl group, DPA can modify body proteins or peptides which leads to creating new epitopes [32]. The appearance of both AChR-Abs and MuSK-Abs (like in our patient) indicates the general autoimmune reaction after DPA [30,31,32,33] as they are rarely both seen in idiopathic MG. The alternative explanation of this phenomenon is that DPA stimulates prostaglandin E1 synthetase, producing prostaglandin E1, which occupies an allosteric site on AChRs, hindering acetylcholine binding [29,30].

DPA is metabolized in the liver and excreted in urine (about 80% as disulfides) over a 48 h period. Only a small amount is removed and excreted within feces. Its elimination is biphasic, initial half-life of about 1 h followed by a lower second phase lasting 5 h. Generally, the excretion heal-life of DPA is 1.7–7 h. However, in long-term treated patients the elimination of DPA can take even 6 days [10].

DPA-inducing autoimmunological reactions may cause autoimmunological disorders like DPA may induce autoimmunological reactions and cause MG as well as lupus-like reactions, so by discontinuing DPA, we eliminated the trigger of this reaction. However, apart from the short half-life of DPA, autoimmunological reactions may persist longer (see: the persistence of MG Abs in MG Table 1), or even in idiopathic disorders DPA may be only the trigger factor of the disease [10].

As the MG phenotype according to AChR-Abs or MuSK-Abs differs, which Abs caused the MG symptoms in our patient can be discussed. In the MuSK-Ab MG group, symptoms onset is observed in young females (third decade) with rapid progression of bulbar involvement in the first stage of disease, neck extensor weakness, higher frequency of myasthenic crisis, lack, and no response to pyridostigmine treatment. However, rarely, ocular manifestations with symmetrical ophthalmoparesis (especially horizontal gaze paresis) and rapid-remitting diplopia may occur. In the AChR-Ab MG group, late onset (>50 years of age) is seen most frequently in men, with variable symptoms, including ocular manifestation (diplopia, symmetrical ptosis) and good response to acetylcholinesterase inhibitors (pyridostigmine) [16,34,35]. Hence, based on these observations, the phenotype of MG in our patient seems to be related to AChR-Abs, but the presence of both Abs supports us to diagnose the general autoimmune reaction caused by DPA treatment [35,36,37,38]. However, in the literature, cases of DPA-induced MG without Abs against AChR and MuSK are also described (up to 20% of cases) [16,18,27].

Further, in our patient, a decrease of Abs levels was observed during follow-up visits, which additionally supported the diagnosis of DPA-induced autoimmune syndrome. However, it was a slow process. Poulas et al. [27] showed that levels of both Abs start to decline slowly one month after DPA cessation; MuSK-Abs disappear completely after four months (like in our patient), but serum AChR-Abs level decline more slowly and in a biphasic manner. Poulas et al. observed an 82% decrease in four months, but complete dissolution of Abs took almost two and a half years.

When a diagnosis of DPA-induced MG is made, treatment includes DPA cessation, which should lead to symptom cessation; however, trientine or zinc salts should be introduced as WD needs lifelong treatment. Anticholinesterase therapy is usually suggested if extraocular symptoms of MG occur (which is very rare), but it mainly depends on symptoms cessation as well as the patient’s and physician’s decisions. Immunosuppressants like steroids are used very rarely, usually in cases when idiopathic MG with severe course is suspected [15,27].

Additionally, it should be mentioned that DPA treatment is associated with increased risk of autoimmune disorders other than MG (Table 2) [9], such as drug-induced lupus, glomerulonephritis, vasculitis, and epidermolysis bullosa acquisita. However, the exact mechanism of these effects is not known. Apart from the etiology of DPA-induced MG described above, it has been proven that DPA may modulate inflammatory processes including reducing the number of T-lymphocytes, triggering the synthesis of autoantibodies and activating macrophage functions since DPA binding to the aldehyde group on macrophages may lead to their activation. Furthermore, this can lead to decreasing serum levels of interleukin-1 and rheumatoid factor, increasing serum levels of interleukin-6, 13, 15, and 23, tumor necrosis factor-alfa, interferon-gamma, and the activation of natural killer cells, which may commonly lead to autoimmunity and autoimmune complications of DPA treatment [9,35,36,37,38,39].

In a study performed by Seessle et al., 2.6% of the analyzed WD patients developed an autoimmune disease during treatment with DPA [36]. However, data about antinuclear antibodies in WD according to the type of treatment are conflicting [35,36].

Discussing our case report, it is also worth mentioning, that despite the favorable clinical outcome in WD-treated patients, the neurological deterioration of clinical symptoms (or de novo neurological symptoms) may occur in up to 11.5% of WD patients [7]. In case of early deterioration (up to 6 months since WD treatment introduction), it is usually related to the type of anti-copper treatment (especially DPA ADR’s) or to the natural course of the disease (usually advanced at diagnosis in such cases) [2,3,7]. “Late” WD neurological deteriorations, diagnosed after 6 months of WD treatment, are usually caused by non-compliance with anti-copper treatment reported mostly by patients’ relations and confirmed by copper metabolism (mainly increased daily urinary copper excretion > 500 µg/24 h with chronic use of chelators or > 100 µg/24 h on zinc salts) [7].

Another cause of neurological symptoms aggravation in WD-treated patients is copper deficiency syndrome (CDS) with neurological symptoms, sensorimotor distal polyneuropathy, spinal cord myelopathy (cervical or thoracic), and rarely, white matter demyelination, epileptic seizures, and hematological changes (especially anemia [sideroblastic] with leukopenia and neutropenia). These syndromes occur as a consequence of WD overtreatment, mostly on zinc salts (with a mean duration of treatment of 15 years). Apart from clinical symptoms, the copper metabolism can let to establish CD diagnosis when daily urinary copper excretion is low: <20 µg/24 h in patients treated with zinc salts or <100 µg/24 h on chelators (including DPA) and associated with low serum total copper and low ceruloplasmin. The cessation of anti-copper treatment, as well as copper substitution, leads to the cessation of CD hematological symptoms; however, neurological symptoms in such cases may sometimes persist [8].

As the de novo neurological symptoms suggesting MG occurred in our patient after 15 months of DPA treatment, the non-compliance with DPA as well as WD overtreatment were excluded based on copper metabolism (urinary copper excretion documented the correct WD treatment)

Finally, it should be mentioned that despite the efficacy of DPA in WD treatment [1,2,3,11,12], several other ADRs besides MG and “paradoxical, early neurological deterioration” [1,9,15] can be caused by DPA. These ADRs can be divided according to the time of onset from DPA introduction as early (up to three weeks after DPA introduction) and late (after more than three weeks) (Table 2) [9]. Symptomatology may sometimes be very severe and physicians treating WD are responsible for their diagnosis [1,2,3]. However, it should be mentioned that apart from the case reports, most of ADRs occur very rarely and their frequency is not established.

## 4. Limitations

As only case reports of patients with post d-penicillamine MG were available in the literature, we were not able to meta-analyze the data. However, this systematic review of the literature may be helpful for practicing physicians and provide overview of available data in this topic. We also did not perform immunological screening according to other auto-immune disorders (Table 2), as there were no, apart from MG clinical symptoms of other DPA-induced disease.

## 5. Conclusions

Neurological deterioration or de novo neurological symptoms in WD patients have to be always widely analyzed and differential diagnosis should include ADRs, non-compliance with anti-copper treatment and WD overtreatment with CD syndrome. In the differential diagnosis process physicians should consider copper metabolism analysis and have the knowledge according to WD course and management [1]. When prescribing treatment for WD, knowledge about the possible ADRs, particularly less-specific effects such as muscular weakness and diplopia (i.e., MG symptoms) is important as they can lead to misdiagnosis, especially in patients with the severe neurological form [7,8,40]. In the case of DPA-induced MG, treatment generally includes DPA cessation and additional treatment of MG symptoms with pyridostigmine. Usually, the response to pyridostigmine treatment is good and MG symptoms remit within one year after DPA treatment cessation. However, sometimes MG symptoms may persist and DPA treatment is only a trigger factor for MG occurrence.

## Figures and Tables

**Figure 1 life-13-01715-f001:**
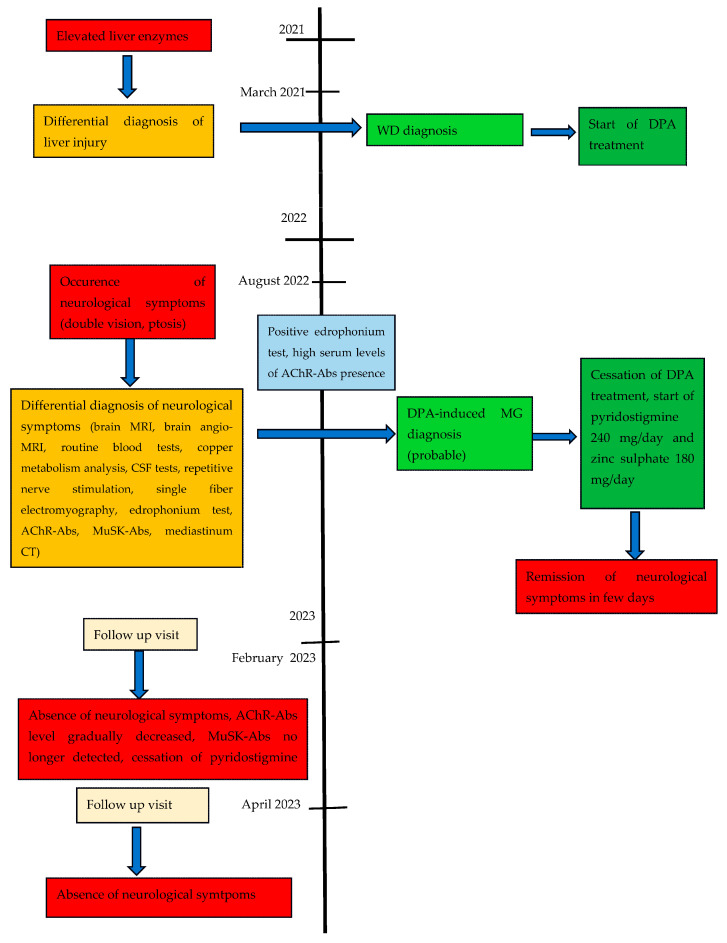
Case report timeline. Presented according to CARE (Case Reports) guidelines. Abbreviations: WD = Wilson’s disease; DPA = d-penicillamine; MG = myasthenia gravis; MRI = magnetic resonance imaging; ACHR Abs = antibodies against human nicotinic acetylcholine receptors; MuSK-Abs: antibodies against muscle-specific kinase (MuSK); CSF = cerebrospinal fluid; CT = computed tomography.

**Figure 2 life-13-01715-f002:**
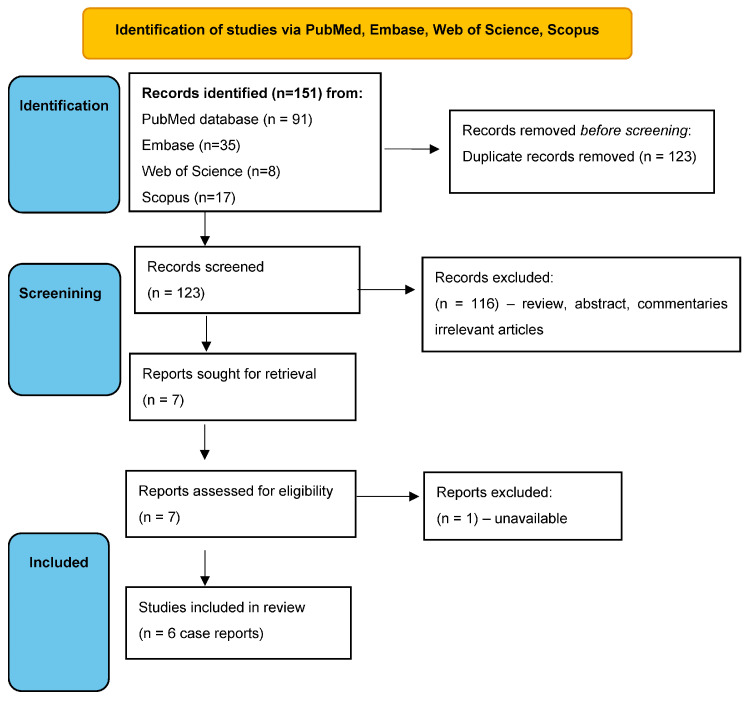
Flow chart of the systematic literature search according to PRISMA guidelines. A total of 151 articles were found during the initial screen and 6 articles were included in the qualitative synthesis.

**Table 1 life-13-01715-t001:** Summary of case reports analyzing DPA-induced myasthenia gravis in patients with Wilson’s disease.

Reference	Patient Characteristics (Age, Gender, WD Symptoms)	Duration, Kind and Dose of WD Treatment	MG Clinical Presentation and Diagnostic Tests	MG Abs Induced by DPA	Treatment and Outcome
Thapa et al. 2022 [20]	15-year-old male with neurological WD (generalized weakness, spastic limbs, hand tremor)	Duration of WD treatment with DPA: 6 years Since diagnosis aged 9, patient was treated with DPA 500 mg/day and zinc 60 mg/day	Bilateral facial weakness (dropping jaw and both upper eyelids) and dysphagia RNS from the facial nerve (orbicularis oculi muscle) showed >10% decremental response Normal chest CT	Positive serum AChR-Abs (14.3 nmol/L; normal < 0.4) Negative for MuSK-Abs	Withdrawal of DPA Pyridostigmine introduced (dose not known) Remarkable improvement within 2 months
Reuner et al. 2019 [21]	17-year-old female with hepatic WD (moderately elevated transaminases)	Duration of WD treatment with DPA: 6 years Since diagnosis aged 11, patient was treated with DPA 300 mg/day with pyridoxine supplementation 20 mg/week	Exercise-induced speech difficulties, mouth and tongue motility and dysphagia RNS showed significant decrement in the orbicularis oris (38%) and trapezius (25%) muscle Chest magnetic resonance imaging visualized thymus hyperplasia	Positive serum AChR-Abs (>5 nM) MuSK-Abs not tested	Continuation of DPA Pyridostigmine introduced (180 mg/day) Robot-assisted endoscopic thymectomy showed marked lymphofolicular hyperplasia (no thymoma) Remarkable improvement, 2 years after cessation of antimyasthenic therapy patient remains asymptomatic
Tan et al. 2012 [22]	16-year-old female with WD (clinically asymptomatic—diagnosed during family screening)	Duration of WD treatment with DPA: 4 months Since diagnosis aged 15 (10 months earlier), patient was treated with DPA 750 mg/day	Diplopia, bilateral ptosis, jaw weakness with difficulties in smiling and chewing, dysarthria and dysphagia, proximal limb weakness (with massive bilateral breast enlargement, galactorrhea, and dermopathy) RNS showed a decremental response of the right deltoid and orbicularis muscles Chest CT not done	AChR-Abs and MuSK-Abs not tested	Withdrawal of DPA; trientine 900 mg/day introduced with clobetasone butyrate for skin lesions (indication dermopathy) One month later disappearance of the visual and skin complaints At follow-up at 7 months, complete resolution of all neurological and dermatological DPA-induced side effects was observed
Varghese et al. 2002 [23]	12-year-old female with hepatic WD (prior hepatomegaly and jaundice)	Duration of WD treatment with DPA: 4 years Since diagnosis aged 8, patient was treated with DPA 1000 mg/day and pyridoxine 1 mg/day	Left-sided ptosis (cranial nerves normal, systemic examination unremarkable) RNS showed decremental response of more than 20% in both deltoid muscles and the left orbicularis oculi Positive edrophonium test Normal chest CT	Positive serum AChR-Abs (29.1 nmol/L; normal < 0.4) MuSK-Abs not tested	Withdrawal of DPA and trientine 1500 mg/day introduced Remarkable improvement within 3 months, all symptoms disappeared After 1 year, AChR-Abs became negative
Masters et al. 1976 [25]	18-year-old female with hepatic WD	Duration of WD treatment with DPA: 8 years Since diagnosis aged 10, patient was treated with DPA 1000 mg/day	Progressive muscular weakness with bilateral ptosis, diplopia, bilateral facial and palatal weakness, dysphagia, dysarthria and marked fatigability RNS showed a decremental response of the right deltoid and orbicularis muscles Positive edrophonium test Normal chest CT	Serum AChR-Abs titer 19 U (normal 1 U) MuSK-Abs not tested	DPA was continued Pyridostigmine (120 mg/day) and neostigmine (15 mg as required) was introduced Thymectomy (enlarged thymus, with thymic hyperplasia) but no thymoma MG symptoms disappeared over 5 months; however, patient was diagnosed as MG and treated with pyridostigmine as increased serum AChR-Abs were observed during follow-up
Czlonkowska et al. 1975 [26]	14-year-old male with WD (clinically asymptomatic—family screening)	Duration of WD treatment with DPA: 13 months Since diagnosis aged 13, patient was treated with DPA 1000 mg/day and pyridoxine 1 mg/day	Initial right-sided ptosis (cranial nerves normal, systemic examination unremarkable). After 10 days, left ptosis also occurred RNS not determined Positive edrophonium test Chest CT not done	Serum AChR-Abs positive MuSK-Abs not tested	Withdrawal of DPA Neostigmine was introduced (dose not known) Remarkable improvement with cessation of all symptoms within 6 weeks After 8 months, DPA 750 mg/day was reintroduced without adverse events during next 10 months

DPA = d-penicillamine; MG = myasthenia gravis; TN = trientine; WD = Wilson’s disease; RNS = repetitive motor nerve stimulation; CT = computed tomography; Abs = antibodies; AChR = human nicotinic acetylcholine receptors; MuSK = muscle-specific tyrosine kinase.

**Table 2 life-13-01715-t002:** Possible adverse drug reactions caused by d-penicillamine, classification according to time of occurrence (early/late), and their estimated frequency [9,10].

Affected System	Symptoms and Their Estimated Frequency	Type of ADR (Early/Late)
Skin	Degenerative dermatoses (elastosis perforans serpiginosa ^a^, cutis laxa (skin laxity) ^a^, anteroderma^x^, pseudo-pseudoxanthoma elasticum ^a^) Bullous dermatoses^x^ (pemphigus and bullous disease) Miscellaneous cutaneous conditions (lichen planus-like eruptions^x^, aphtous stomatitis or glossitis ^a^, oral ulcerations ^a^, alopecia ^a^, psoriasiform dermatitis^x^, seborrheic dermatitis-like picture^x^, yellow-nail syndrome^x^)	late early late
Nervous system	Paradoxical neurological deterioration ^b^, myasthenia-like syndromes^x^, peripheral sensory-motor neuropathies^x^, optic nerve neuropathy^x^, serous retinitis^x^, hypogeusia (diminution in taste perception)^x^, deafness^x^	early/late late
Connective tissue disease	Lupus-like syndrome, rheumatoid arthritis^x^, polymyositis^x^, arthralgia^x^,	late early/late
Renal	Proteinuria ^b^, hematuria ^a^, Goodpasture syndrome^x^, severe fatal glomerulonephritis associated with intra-alveolar hemorrhage^x^, renal vasculitis^x^	early/late late
Respiratory	Bronchiolitis^x^, pulmonary fibrosis^x^, pneumonitis^x^, pleural effusion^x^, dyspnea^x^	late
Gastrointestinal	Nausea^x^, vomiting^x^, diarrhea^x^, cholestatic jaundice^x^, liver siderosis^x^	early/late late
Hematologic	Thrombocytopenia ^c^, neutropenia^x^, hemolytic anemia^x^ Agranulocytosis^x^, aplastic anemia^x^,	early/late late
Immunologic	Immunoglobulin deficiency^x^	late
Reproductive system and breast disorders	Breast enlargement ^a^	late

^a^ rare (between 1/1000 and 1/10,000; ^b^ common (between 1/10 and 1/1000); ^c^ very common (>1/10).

## Data Availability

All of the data are available upon request to the corresponding author.

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
