# Peer review of "D-Penicillamine-Induced Myasthenia Gravis—A Probable Complication of Wilson’s Disease Treatment—A Case Report and Systematic Review of the Literature"

_life, 2023, doi:10.3390/life13081715_

Round 1
Reviewer 1 Report
This article reports on a case of myasthenia gravis induced by treatment with D-penicillamine in a patient with Wilson's disease. The authors concisely present case details and provide a comprehensive review of the relevant literature. The background knowledge of the disease is discussed thoroughly in the text, providing the readers with a comprehensive understanding. Overall, this case report and literature review provide valuable clinical knowledge on the disease.
I have a few questions and suggestions for the article:
1. Table 1 might add columns for the dose and duration of WD treatment, the Abs of the induced MG, and the post-treatment outcome, to provide a more precise and concise clinical picture of the disease.
2. The reported case showed double autoantibodies. Double positive MG is rare and should be an important feature of this case, as most anti-MuSK antibodies, unlike Ach antibodies, belong to the IgG4 immunoglobulin subclass, which is incapable of activating complement or inducing antigenic modulation. Thus, as the authors state, this suggests that DPA-induced myasthenia gravis may act on broader or more generalized immune mechanisms than "typical" MG. A high percentage of myasthenia gravis patients also have other autoimmune abnormalities. Has this patient been tested for other autoimmune antibodies?
3. Can the authors provide the Mean Consecutive Difference of this patient's single-fiber EMG? Were the electrophysiologic studies repeated after the MG symptoms resolved?
Finally, I would like to thank the author and editor for allowing me to review this article and for the opportunity to learn from this case report.
Author Response
RESPONSE TO THE REVIEWERS
Reviewer 1
Thank you very much for your review and comments. Below, we responded to the concerns that you have raised.
Comment:
This article reports on a case of myasthenia gravis induced by treatment with D-penicillamine in a patient with Wilson's disease. The authors concisely present case details and provide a comprehensive review of the relevant literature. The background knowledge of the disease is discussed thoroughly in the text, providing the readers with a comprehensive understanding. Overall, this case report and literature review provide valuable clinical knowledge on the disease.
I have a few questions and suggestions for the article:
Comment 1
Table 1 might add columns for the dose and duration of WD treatment, the Abs of the induced MG, and the post-treatment outcome, to provide a more precise and concise clinical picture of the disease.
Answer:
Thank You for the comment. We added columns for dose and duration of DPA treatment, Abs and outcome as proposed.
Comment: 2.
The reported case showed double autoantibodies. Double positive MG is rare and should be an important feature of this case, as most anti-MuSK antibodies, unlike Ach antibodies, belong to the IgG4 immunoglobulin subclass, which is incapable of activating complement or inducing antigenic modulation. Thus, as the authors state, this suggests that DPA-induced myasthenia gravis may act on broader or more generalized immune mechanisms than "typical" MG. A high percentage of myasthenia gravis patients also have other autoimmune abnormalities. Has this patient been tested for other autoimmune antibodies?
Answer:
Thank You for the comment. We didn’t perform analysis of other auto-antibodies. Apart from MG symptoms patient had no other clinical symptoms. The diagnosis of MG caused by DPA in WD was established. After DPA cessation MG symptoms resolved. . Hence, we didn’t perform further auto-immunological diagnostics (apart from MG antibodies).
We added it in study limitation chapter.
Comment 3. Can the authors provide the Mean Consecutive Difference of this patient's single-fiber EMG? Were the electrophysiologic studies repeated after the MG symptoms resolved?
Answer:
Thank You for the comment:
At follow up visit after 6 months: the Mean Consecutive Difference of this patient's single-fiber EMG from orbicularis oculi muscle repeated after the MG symptoms resolved was 23 µs (laboratory norm up to 31 µs) (we added it in the text).
Finally, I would like to thank the author and editor for allowing me to review this article and for the opportunity to learn from this case report.
Thank You very much for review.

Reviewer 2 Report
Dear Author(s),
Thank you for your manuscript. Unfortunately, I do not think that your case report is as innovative to be published in a prestigious journal such as Life in the present form; modification is needed...
If I were you I would reach for a journal with lower Q and IF, since this is only a case report with somewhat of a literature review. However, if you want to publish it within Life, then all of the modifications are necessary:
The introduction is too extensive, there is much unnecessary info.
At the end of the introduction aim needs to be mentioned.
Case description is lacking. Since you are describing potential AE, there must be important info like overall Naranjo scale (please include individual determinants) etc., from the clinical pharmacology and pharmacovigilance point of view. What is more, a flow chart/case time-line would be useful for easier follow-up of the case.
Methodological search is lacking. Including only PubMed can be misleading due to low coverage.
Potential pathophysiological background of AE needs to be explained more extensively within discussion section.
Please comment of reversibility of AE. Is the half-life (4-5X) of D-penicillamine an important variable for resolution; provide elaboration and comment on relevance of half-life, if applicable?
Please include approximate rate of occurrence for AE mentioned in Table 2.
Limitations and strengths needs to be mentioned within discussion section.
It would be great if you can visually present (as figure or graphical abstract) recommendations what to do when D-penicillamine-induced myasthenia gravis happens -> clinical flow-chart with recommendations for diagnosis, management and follow-up
Best regards, Reviewer
Author Response
RESPONSE TO THE REVIEWERS
Reviewer 2
Thank you very much for your review and comments. Below, we responded to the concerns that you have raised.
Reviewer 2:
Thank you for your manuscript. Unfortunately, I do not think that your case report is as innovative to be published in a prestigious journal such as Life in the present form; modification is needed...
If I were you I would reach for a journal with lower Q and IF, since this is only a case report with somewhat of a literature review. However, if you want to publish it within Life, then all of the modifications are necessary:
Comment 1;
The introduction is too extensive, there is much unnecessary info. At the end of the introduction aim needs to be mentioned.
Answer:
Thank You for the comment, we shortened the introduction as suggested. For physicians involved in WD any new neurological symptoms or neurological deterioration is an indication for a quick search for the cause of deterioration. Causes may be different, in some WD patients copper deficiency may occur, in others neurological deterioration (especially on DPA), in some ADRs like MG may occur, which should be quickly diagnosed as WD overtreatment requires discontinuation of the anti-copper treatment, neurological deterioration usually require the change of anti-copper drugs, and ADRS related to DPA like MG should lead to quick DPA discontinuation and MG symptomatic treatment – we provided this information it in the revised text (in introduction and discussion sections), as well as in aims of the study
We also defined the aims of the study.
“Since WD patients may present with a wide spectrum of neurological symptoms, diagnosing DPA-induced MG may be difficult. Additionally, any neurological deterioration in WD needs to be quickly diagnosed and treated according to leading pathology (DPA related deterioration? non-compliance with anti-copper drugs? WD overtreatment?) [7-8]. As there is little data in the literature about DPA-induced MG, we aimed to present a case of a patient with WD who developed DPA-induced MG and whose symptoms completely subsided after DPA cessation, with an additional systematic review of DPA-induced MG in WD”.
Comment 2:
Case description is lacking. Since you are describing potential AE, there must be important info like overall Naranjo scale (please include individual determinants) etc., from the clinical pharmacology and pharmacovigilance point of view. What is more, a flow chart/case time-line would be useful for easier follow-up of the case.
Answer:
Thank You for the comment, we added the Naranjo score and described the differential diagnosis of neurological deterioration in our patient which included copper deficiency syndrome as well as non-compliance with DPA. We analyzed copper metabolism in our patient during follow-up providing this data, as well as brain MRI.
Comment 3:
Methodological search is lacking. Including only PubMed can be misleading due to low coverage.
Answer:
Thank You for this valuable comment. We extended our search using Web of Science, SCOPUS and Embase, however we did not find more case reports according to MG in WD induced with DPA. However, we modified the methodology of the article according to this search.
Comment 4:
Potential pathophysiological background of AE needs to be explained more extensively within discussion section.
Please comment of reversibility of AE. Is the half-life (4-5X) of D-penicillamine an important variable for resolution; provide elaboration and comment on relevance of half-life, if applicable?
Answer:
DPA is metabolized in liver, primarily in a phase II reactions (that is conjugated with a sulfide or methylated), the excreted in urine (about 80% a disulfides) over a 48-hours period. A small amount of DPA is excreted with feces. The elimination of DPA is biphasic, with an initial half-life of about 1 hour, followed by a lower second phase of 5 hours (generally, excretion half-life is 1.7-7hours). However there are reports that during chronic DPA treatment, elimination can take as long as 6 days.
DPA may induce autoimmunological reactions, cause MG or lupus-like reactions, so by discontinuation of DPA we eliminated the trigger of this reaction. However, auto-antibodies as well inflammatory reaction may persist longer in case of DPA induced disease (in cases described Abs may persists for even up to one year), or DPA can be only the trigger of autoimmunological disease.
We added this information in discussion.
The added text is as follows:
“DPA is metabolized in the liver and excreted in urine (about 80% as disulfides) over a 48-hours period. Only small amount is excreted with feces. Its elimination is biphasic, initial half-life of about 1 hour followed by a second phase lasting 5 hours. Generally, the excretion heal-life of DPA is 1.7-7 hours. However, in long-term treated patients the elimination of DPA can take even 6 days [10].
DPA may induce inducing the autoimmunological reactions, cause MG as well as or lupus-like reactions, so by discontinuing of DPA we removed eliminated the trigger of this reaction. However, apart from the short half-life of DPA, autoimmunological reactions may persist longer (see: the persistence of MG Abs in MG Table 1), or even in idiopathic disorders DPA may be only the trigger factor of the disease [10]”.
We didn’t try to introduce again the DPA treatment in our patients, hence we didn’t analyze the re-occurrence of WD after DPA in this patient.
Comment 5:
Please include approximate rate of occurrence for AE mentioned in Table 2.
Answer:
We added the estimated frequency of each ADRs in Table 2 (based on literature, however exact data are limited).
Comment 6:
Limitations and strengths needs to be mentioned within discussion section.
Answer:
We added the study limitations in the text, additionally we emphasized and discussed more widely the significance of our case.
It included differential diagnosis of neurological deterioration (or de novo occurrence of neurological symptoms) like in our case as well as knowledge according to ADR’s which can occur during WD treatment. We think that such knowledge is very important for physicians involved in WD treatment, as the wrong diagnosis of MG induced by DPA in WD may lead to intensification of anti-copper treatment (by increasing the doses of DPA) and more severe MG symptoms. Moreover incorrect diagnosis of copper deficiency syndrome may lead to increasing the doses of anti-copper treatment and irreversible neurological and hematological deterioration.
Our case based on new neurological symptoms as a result of WD treated, show difficulties and pitfalls in WD treatment.
Comment 7:
It would be great if you can visually present (as figure or graphical abstract) recommendations what to do when D-penicillamine-induced myasthenia gravis happens -> clinical flow-chart with recommendations for diagnosis, management and follow-up
Answer:
We added graphical abstract including the differential diagnosis of WD neurological deterioration including MG.
Thank You very much for review.

Round 2
Reviewer 2 Report
The quality got improved significantly after modifications as per my comments.
You should still need to include flow chart/case time-line. Please check CARE guidelines for case-reports and do it accordingly.
Best regards, Reviewer
Author Response
Reviewer 2:
Comment:
You should still need to include flow chart/case time-line. Please check CARE guidelines for case-reports and do it accordingly.
Best regards, Reviewer
Answer:
Thank You very much for review, we added as suggested Figure 1 Case report timeline. Presented according to CARE (Case Reports) guidelines.
Thank You very much for review.
